# Anaplastic Mammary Carcinoma in Cat

**DOI:** 10.3390/vetsci8050077

**Published:** 2021-05-04

**Authors:** Maria Soares, Jorge Correia, Catarina Nascimento, Fernando Ferreira

**Affiliations:** CIISA, Faculdade de Medicina Veterinária, Universidade de Lisboa, Av. da Universidade Técnica, 1300-477 Lisboa, Portugal; maria.soares@ulusofona.pt (M.S.); jcorreia@fmv.ulisboa.pt (J.C.); catnasc@fmv.ulisboa.pt (C.N.)

**Keywords:** anaplastic, feline mammary carcinoma, clinicopathological, immunohistochemical

## Abstract

Clinical, pathological, and immunohistochemical findings related to a feline mammary tumor with similar features to canine anaplastic mammary carcinoma are herein described for the first time. A female cat was presented for clinical evaluation with gastrointestinal signs, oedema, erythema, and painful lesion in the right inguinal region. Three weeks later, the mass had doubled in size and radiographic revaluation of the thoracic cavity revealed a metastatic pattern. Due to the poor prognosis and decline of the clinical status the owners decided for euthanasia. Post-mortem examination exposed a mammary tumoral mass with subcutaneous oedema, an enlargement of the right inguinal lymph node, and nodules in several organs. Histological analysis confirmed the presence of large pleomorphic epithelial cells, often grouped in small clusters with bizarre nuclei. Immunohistochemical study of the different lesions was performed and both primary tumor and regional metastasis showed tumor cells to be negative estrogen receptor alpha, positive progesterone receptor, positive HER-2, and positive pan-cytokeratin. Given that the clinical history was compatible with an inflammatory mammary carcinoma, the cyclooxygenase-2 expression levels were evaluated and presented a weak immunoreactivity. Regarding the distant metastatic lesions, tumor cells were negative for ER-α and PR and, positive both for HER-2 and pan-cytokeratin.

## 1. Introduction

Feline mammary carcinoma shows a high incidence, being the third most common tumor in this species [1]. Notwithstanding, feline anaplastic mammary carcinoma (AMC) was only briefly described by Della Salda et al. (1993) [2] and so far, no additional reports were published.

In dogs, the AMC is considered the most aggressive type of mammary tumor, showing a low frequency (4%) and being associated with a short survival time (less than 20 weeks) [3,4,5]. This histologic presentation is characterized by a diffuse infiltrating neoplasm composed of large, pleomorphic and multinucleated giant cells, frequently showing acidophilic cytoplasm and dispersed nuclear chromatin [2]. Lymphocytes, plasm cells, mast cells, neutrophils, eosinophils and macrophages may be presented among the tumor cells and in the stroma [6].

To the best of our knowledge, this is the first report that fully describes the clinical and molecular features of an AMC in a female cat.

## 2. Case Presentation

A 9-year-old spayed female cat of the European shorthair breed with vaccination and deworming overdue was presented at the Teaching Hospital of the Veterinary Medicine Faculty of Lisbon University with signs of vomit and melena. Physical examination revealed a non-pruritic, small nodule (<1 cm), surrounding the nipple of the inguinal right mammary gland. Regional pain was noted during palpation, and the subcutis was firm. The mass was subjected to a fine needle aspiration biopsy for cytological study and a mammary carcinoma was diagnosed based on the presence of cellular aggregates with neoplastic features, such as pleomorphic cells with bizarre nuclei and multinucleated cells, suggesting a high degree of malignancy (Figure 1).

Disease staging was based on the radiological evaluation of the thoracic cavity, with no signs of lung metastases, and hematological and biochemical profiles, neither presenting abnormal values. A radical mastectomy was then advised. In a period of three weeks, the clinical status of the animal rapidly deteriorated and the tumoral mass grew exuberantly (3 cm in diameter). Surgery could not be performed because animal developed anorexia, dehydration, tachypnea, abdominal distension (secondary to aerophagia due to the dyspnea), a weak femoral artery pulse, and lymphadenopathy of both the axillary and inguinal lymph regions. A systemic inflammatory reaction was diagnosed due to leucocytosis (31,400 cells/µL, reference values: 5500–19,500 cells/µL), and neutrophilia (28,260 cells/µL, reference values 2500–12,500 cells/µL) [7]. A radiographic thoracic re-evaluation showed a nodular interstitial lung pattern suggestive of a recent metastatic process. The abdominal ultrasound exam revealed enlargement of both mesenteric and colic lymph nodes (7 mm in diameter), although a normal biochemical profile was maintained.

Given the rapid evolution of a neoplastic disease of poor prognosis and attending the owners’ will, the animal was humanely euthanized. A thorough necropsy was performed, with subsequent histopathological analysis of various organs and tissues.

The following gross modifications were observed: (1) thickening of the skin with underlying subcutaneous oedema in the right inguinal area, (2) nonencapsulated mass with 3 cm in diameter at the right inguinal mammary gland, confined to the subcutaneous tissue (hypodermis), without showing dermic infiltration or into the underlying skeletal muscle (microscopically, the mass showed three smaller nodular lesions), and (3) homolateral inguinal and axillary lymph node enlargement. Moderate hemothorax and hemopericardium were observed, while scattered multifocal, small, nodular lesions could be observed throughout the lung parenchyma (and had a diameter of approximately 1 mm). Multiple nodular-shaped lesions (with a diameter of approximately 3–6 mm) were also observed in the liver, pancreas, right kidney, adrenal glands, cerebellum, brain, and several muscles. No other significant changes could be detected in the remaining organs.

## 3. Material and Methods

Tissue samples from several of the aforementioned lesions were collected and fixed in 10% buffered formalin for 24 h prior to routine histological processing. Histological examination of the lesions, including the mammary mass from which the cytological sample had been taken, revealed an epithelial malignant tumor. The primary tumor corresponded to a round mass of epithelial cells arranged in sheets separated by wide trabeculae of connective tissue, displaying a strong vascular invasion by neoplastic cells (Figure 2A). Extensive necrotic areas were visible (Figure 2B). Neoplastic cells were frequently individualized or arranged in small nests (Figure 2C), often surrounded by collagen fibers (desmoplasia). Small and rare areas of tubular differentiation can be identified. The predominant tumor cells are round or deformed due to close contact. The cytoplasm is acidophilic, and the large nucleus is very often deformed, hyperchromatic, with the nucleolus rarely identified. Giant cells are frequent, both macro and multinucleated (Figure 2D). The metastases (Figure 2E,F) in general showed cellular organization and morphology similar to the primitive tumor, with only small areas of tubular differentiation. According to Misdorp grade system this tumor was classified as highly malignant mammary carcinoma (grade III) [8].

Considering that tumor cells were poorly differentiated and that both anaplastic carcinomas and anaplastic sarcomas display this feature, pan-cytokeratin immunostaining was performed to verify the epithelial nature of the tumor and to help discriminate between these tumor types [9]. Moreover, the proliferation rate of the tumor was determined by calculating the Ki-67 index. Accordingly, the Ki-67 proliferation index was assessed by analyzing the percentage of Ki-67-positive tumor cell nuclei in 1000 tumor cells [10]. For molecular tumor classification, the expression levels of estrogen receptor alpha (ER-α) and the progesterone receptor (PR) were examined in addition to the HER-2 transmembrane receptor status. The antibodies used in this study were previously tested in feline mammary tissues [11,12], being discriminated in Table 1 with the immunohistochemistry protocols. Positive and negative controls were used for each antigen evaluated: a normal tonsil tissue sample for pan-cytokeratin and Ki-67 [10,13]; human mammary carcinomas tissue sample with known positive/negative status for the estrogen receptor alpha (ER-α), progesterone receptor (PR) and HER2 assessment [14], and feline uterus with cystic endometrial hyperplasia-pyometra complex positive for cyclooxygenase-2 (COX-2) [15].

## 4. Results

Strong staining for pan-cytokeratin in the cell membrane confirmed the epithelial nature of the tumor cells (Figure 3A), while a 0.19 Ki-67 value indicated a fast rate of cell growth and confirmed the aggressive nature of the tumor [16] (Figure 3B).

Regarding the immunohistochemistry results, the primary tumor and the regional metastatic lesions were negative for ER-α and positive for PR (Figure 4A,B), while the distant metastasis (pancreas, lung, right kidney and right suprascapular muscle) were negative for both hormone receptors (Figure 4C), according to the American Society of Clinical Oncology (ASCO) guidelines [17]. Concerning HER-2 expression, primary and metastatic lesions showed a strong and complete membrane staining in more than 30% of tumor cells (Figure 4D), scored as 2+ in accordance with the guidelines of the ASCO [18]. The COX-2 expression in the primary mass was scored as 1+ (cytoplasmic staining in < 10% of tumor cells) corresponding to a weak immunoreactivity [14].

## 5. Discussion

Based on the clinical evolution and the histopathological exam, the tumor herein described was considered compatible with an anaplastic mammary carcinoma (AMC). In the dog, its main features include high infiltrative capacity associated with frequent lymphatic vessels invasion, leading to regional lymph node and lung metastasis [19,20], with radiographic evaluation usually detecting an interstitial lung pattern [19], as reported in this case. Moreover, the AMC type is characterized by the presence of multinucleated cells together with large, pleomorphic cells with bizarre nuclei varying in size and usually chromatin enriched. Both anisokaryosis and anisocytosis were observed, and mitoses were common events (8–10 dividing-cells were observed per each high magnification field, 40×). Frequently, the neoplastic cells are both presented individually and grouped in small nests and the tumor tissues can be infiltrated by lymphocytes, plasma cells, mast cells, neutrophils, eosinophils and macrophages [19,20,21]. All the above anaplastic mammary carcinoma features are very similar to those reported in the presented case. Additionally, the tumor was poorly differentiated, which is frequently observed in this type of neoplasm and often requires the evaluation of protein markers, like cytokeratins, to prove the epithelial cellular origin.

In human oncology, determination of ER-α, PR and HER2 status is routinely performed, and has an important prognostic and predictive value together with the Ki-67 index. The expression of ER-α and PR are usually associated with a better prognosis, and patients whose biopsies show positive staining for these receptors are eligible for hormonal therapy. In contrast, human breast HER2-positive tumors, usually negative for ER-α and PR expressions are correlated with a worse prognosis, and have a specific target therapy with an anti-HER2 monoclonal antibody [22]. In veterinary oncology, the significance of this classification is not well defined, but HER2 status has been associated with a shorter overall survival lifespan in feline mammary carcinoma cases [23]. Regarding the presented case and considering the high cellular proliferation index and the HER-2 overexpression, this tumor also showed morphologic and immunohistochemical characteristics usually associated with an aggressive tumor behavior with poor outcome [22,23]. Moreover, we have identified a different molecular pattern in the tumor cells of the distant metastases, namely a loss of PR staining, which is also reported in human cases as being correlated with a less favorable prognosis [2]. Accordingly, previous studies in canine and feline medicine described a progressive down expression of hormone receptors in metastases [24,25,26].

Finally, the tumor described in this case report could have been classified as inflammatory, like mammary carcinoma because of the clinical presentation, such as the fulminant clinical course, the oedema, erythema, firmness, and hyperthermia of the mammary glands. However, the inflammatory carcinoma differs from other carcinomas due to repeated tumor embolization in the lymphatic vessels considered a specific hallmark of this mammary tumor type [27,28]. In fact, massive embolization of neoplastic cells in the dermal lymphatic vessels was not observed. Additionally, it was considered relevant to analyze the expression of COX-2 (that showed a 1+ score), a feature that needs to be evaluated in future AMC studies.

## 6. Conclusions

In our opinion, this report emphasizes the importance of using both histopathological and molecular markers to correctly establish the diagnosis and the prognosis, providing new information concerning feline mammary carcinomas. Although feline AMC is rare, we consider that the increase in the longevity will lead to higher incidence of this tumor type. Therefore, clinical veterinarians must be aware of the possible occurrence of this tumor, suspecting it in cases of neoplasms with extremely rapid growth and with evident signs of high malignancy, such as skin ulceration, adhesion to surrounded tissues (skin and muscle) and development of depressed central zone due to tumor necrosis. In case of suspecting an AMC, the clinical veterinarian should start the treatment urgently and perform an adequate chemotherapy protocol for this aggressive cancer.

## Figures and Tables

**Figure 1 vetsci-08-00077-f001:**
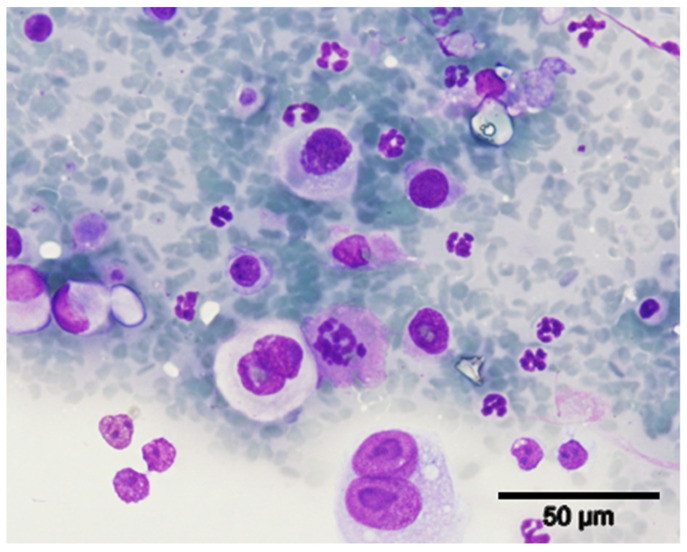
Fine needle aspiration cytology of the right inguinal mammary lesion. The tumor cells were frequently multinucleated and showed marked anisocytosis, anisokaryosis and multiple prominent nucleoli, indicating a high degree of malignancy. Inflammatory cells, particularly neutrophils and lymphocytes, were frequently observed (Giemsa counterstain, 40×).

**Figure 2 vetsci-08-00077-f002:**
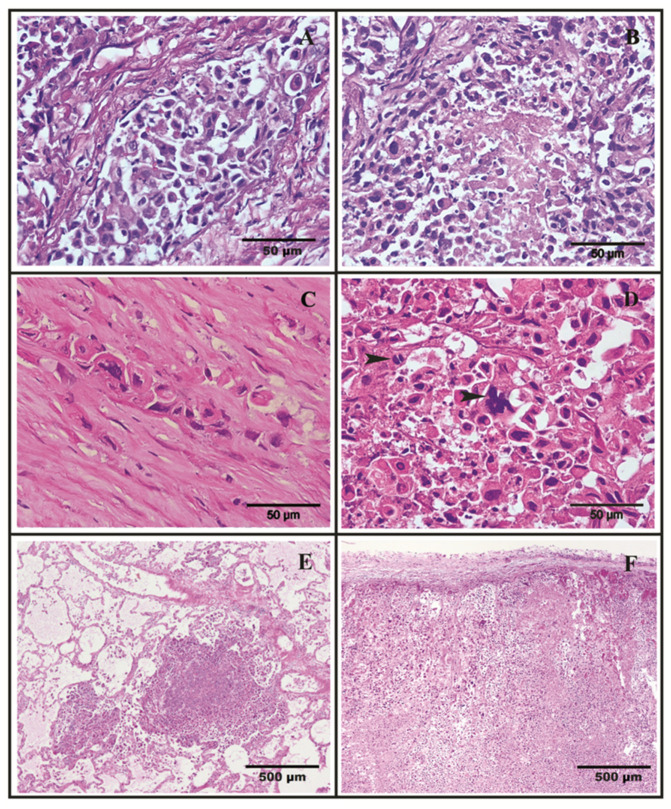
(**A**) Primary mammary tumor tissue with vascular invasion by neoplastic cells (H&E staining, 40×); (**B**) Necrosis affecting large areas of the tumor (H&E, 40×); (**C**) Neoplastic cells are frequently individualized or arranged in small nests surrounded by dense and abundant collagen fibers (desmoplasia) (H&E staining, 40×); (**D**) Giant and multinucleated cells (arrowheads) were frequently present, and severe anisokaryosis and anisocytosis were common. The nuclei were irregularly shaped with occasional indentations or convolutions present and had coarsely stippled chromatin (H&E, 40×). (**E**) Lung and (**F**) supra renal metastases of primary mammary tumor (H&E staining, 4×).

**Figure 3 vetsci-08-00077-f003:**
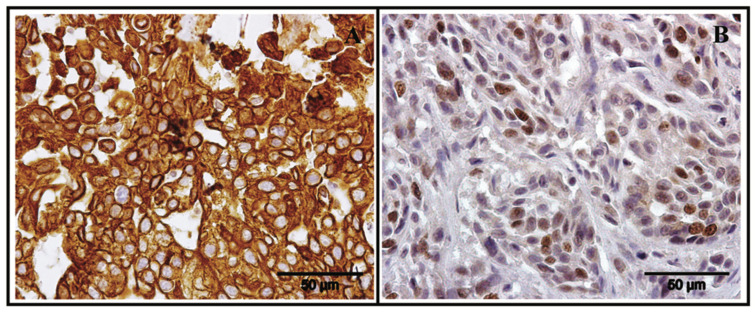
(**A**) Intense membrane pan-cytokeratin immunostaining in the primary mammary mass (chromogen diaminobenzidine, Mayer’s hematoxylin counterstain, 40× magnification); (**B**) Nuclear Ki-67 immunostaining (0.19 index) (chromogen diaminobenzidine, Mayer’s hematoxylin counterstain, 40×).

**Figure 4 vetsci-08-00077-f004:**
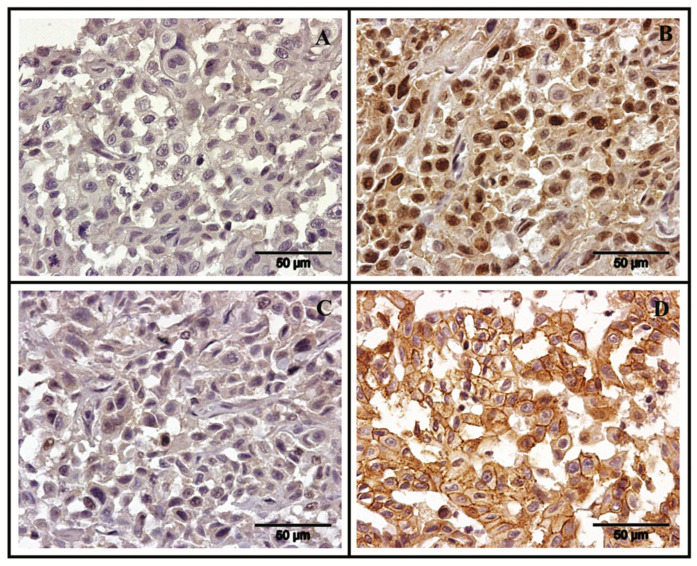
Primary and metastatic tumor immunophenotyping (chromogen diaminobenzidine, Mayer’s hematoxylin counterstain, 40×). (**A**) Primary tumor stained negative for ER; (**B**) PR positive staining was observed in more than 10% of tumor cells present in a metastatic lesion of the regional inguinal lymph node; (**C**) Negative immunostaining for PR expression in distant metastases (suprascapular muscle); (**D**) HER-2 positive immunostaining (scored 2+) is shown. A moderate, complete membrane staining pattern of non-uniform intensity was observed in more than 10% of the tumor cells in the primary tumor.

**Table 1 vetsci-08-00077-t001:** Primary antibodies and immunostaining protocols used.

Antigen	Primary Antibody	Dilution	Incubation	Antigen Retrieval Method
Cytokeratins	clone AE1/AE3 (DAKO)	1:100	60 min, RT	Microwave 900 W 5′ + 600 W 15′ with buffer solution at pH 9,0 (Tris EDTA pH 9.0, Novocastra™ Epitope Retrieval Solution)
Ki-67	clone MM1 (Leica)	1:100	60 min, RT	Pressure chamber (2 atm during 2 min), citrate buffer solution (pH 6.0)
ER-α	clone SP1 (Ventana)	1:150	16 h, 4 °C	
HER-2	clone A0485 (DAKO)	1:300	60 min, RT	Boiling water bath (95 °C, 60 min), citrate buffer solution (pH 6.0)
PR	clone 1E2 (Ventana)	RTU	16 h, 4 °C	
COX-2	clone SP21 (Biocare Medical)	1:75	16 h, 4 °C	Pressure chamber (2 atm during 2 min), citrate buffer solution (pH 6.0)

RT, Room temperature; RTU, ready-to-use.

## Data Availability

Data sharing not applicable.

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
