# Peer review of "Anaplastic Mammary Carcinoma in Cat"

_vetsci, 2021, doi:10.3390/vetsci8050077_

Round 1
Reviewer 1 Report
the case report submitted by Maria Soares is interesting,
however, I have some suggestions.
1- in the introduction part, could the author explain better what the AMC is, and what is important for the diagnosis and the prognosis. could the authors add some more references.
2 in lines 56 and 57, can the authors add the references of the references values.
3- I prefer that the case report be divided into several separated parts; so can the author split the results from the "case presentation" and separate the materials and methods part from the results. In this way, The case report is easier for the reader to understand.
4- in line 104 the author writes " by calculating the Ki-67 index". can the author explain how they calculated this index and add the references?
5- in line 111 write the state, like in line 108
6- from line 106 the author describes the positive tissues: add the reference from each positive sample
Author Response
Reviewer #1 (Comments to the Author):
The case report submitted by Maria Soares is interesting, however, I have some suggestions. Dear reviewer, thank you for your comments and suggestions. All the authors appreciate your valuable review.
1- in the introduction part, could the author explain better what the AMC is, and what is important for the diagnosis and the prognosis. could the authors add some more references. Dear reviewer, further information regarding the AMC was added in lines 29-32 and 35-39, and three references were added.
2- in lines 56 and 57, can the authors add the references of the references values. As suggested, a new reference was added in line 69.
3- I prefer that the case report be divided into several separated parts; so can the author split the results from the "case presentation" and separate the materials and methods part from the results. In this way, The case report is easier for the reader to understand. As requested, the case report was divided into Introduction, Case Presentation, Material and Methods, Results, Discussion, and Conclusion.
4- in line 104 the author writes " by calculating the Ki-67 index". can the author explain how they calculated this index and add the references? Dear reviewer, additional information concerning the analysis of Ki-67 index was added in lines 120 and 121, as well as a reference.
5- in line 111 write the state, like in line 108. Dear reviewer, since revisor 2 suggested deleting the description of all the antibodies from the main text, once the information is repeated in table 1, the data from antibodies were removed.
6- from line 106 the author describes the positive tissues: add the reference from each positive sample. As requested, four references were added in lines 133, 135 and 136.
Reviewer 2 Report
I carefully reviewed the manuscript entitled “Anaplastic feline mammary carcinoma: The first clinicopathological and immunohistochemical description” by Soares and colleagues. This is an interesting and well developed case report of a very rare type of feline mammary tumor. Even if it is not the first case of anaplastic mammary cancer in cats, the authors’ approach allowed a complete description and classification of this condition. In my opinion, the manuscript needs only some minor revisions before to be considered suitable for the publication in the journal.
Here below the point that should be amended:
- This is not the first case in literature of an anaplastic carcinoma in cats. The first description was performed in 1993 by Della Salda and colleagues. Anyway, the approach used in the present study is very complete and it adds new information and a new case of this condition, giving an important contribution for the feline oncology. The authors should modify the title and the lines 30-31 of the introduction in order to specify that it is not the first case, but the first molecular characterization of this cancer in cats.
- In lines 178-181, when discussing the discordance between the molecular profile of the primary tumors and its metastases, the authors only cited human reports to explain this finding. In the last 10 years, some important literature has been produced also in canine and feline medicine ((Brunetti et al, 2013; Beha et al, 2012 and 2014), with similar findings that should be integrated to the discussion.
- Line 33: if it possible, the age of the sterilization should be reported, since it could add an interesting data, in particular concerning the reactivity to ER and PR IHC.
- To confirm what stated in lines 80-81, the authors should add a photo of the anti-CK IHC of vessels, to show that the cells are actually neoplastic epithelial cells. If not possible, the authors should provide a photo of HE at higher magnification.
- It could be interesting to have also a photo of the HE of distant metastases.
- Line 103: please clarify which method the authors used to count the Ki67 index, since in literature there are several methods.
- I suggest to delete the part concerning the description of all the antibodies used in the study, since the table is very clear and this part is thus redundant.
- An important point, lines 188-190. The authors stated that the exclusion from the differential diagnosis of the inflammatory carcinoma was also based on the COX2 weak staining. Based on my experience in COX2 and mammary cancer studies, I think that there is a common misunderstanding about the inflammatory pathogenesis of the inflammatory carcinoma, also due to its name. actually, this name comes from the macroscopic aspect (caused by lymphatic vessels massive embolization), and not to any concurrent inflammation. For this reason, even if previous articles showed a strong positivity to anti-COX2 IHC, it does not mean that COX2 is a marker of inflammatory cancer. In my opinion, this finding is not needed to discriminate between and anaplastic from an inflammatory carcinoma, I think that what you have described about macro and microscopic findings is sufficient to support your diagnosis.
- Conclusion: the authors built this section only on the utility of the histological and IHC analyses in diagnose this disease. I suggest to emphasize also the diagnosis in its own, since it concerns a very rare (for the moment) and aggressive cancer in the feline species.
Author Response
Reviewer #2 (Comments to the Author):
I carefully reviewed the manuscript entitled “Anaplastic feline mammary carcinoma: The first clinicopathological and immunohistochemical description” by Soares and colleagues. This is an interesting and well developed case report of a very rare type of feline mammary tumor. Even if it is not the first case of anaplastic mammary cancer in cats, the authors’ approach allowed a complete description and classification of this condition. In my opinion, the manuscript needs only some minor revisions before to be considered suitable for the publication in the journal. Dear reviewer, thank you very much for your insightful comments and constructive remarks along with this case report. They really improve the final quality of this manuscript.
Here below the point that should be amended:
This is not the first case in the literature of an anaplastic carcinoma in cats. The first description was performed in 1993 by Della Salda and colleagues. Anyway, the approach used in the present study is very complete and it adds new information and a new case of this condition, giving an important contribution for feline oncology. The authors should modify the title and the lines 30-31 of the introduction in order to specify that it is not the first case, but the first molecular characterization of this cancer in cats. Dear reviewer, thank you very much for this relevant remark. Accordingly, the title and lines 40-41 of the introduction were modified.
In lines 178-181, when discussing the discordance between the molecular profile of the primary tumors and its metastases, the authors only cited human reports to explain this finding. In the last 10 years, some important literature has been produced also in canine and feline medicine ((Brunetti et al, 2013; Beha et al, 2012 and 2014), with similar findings that should be integrated to the discussion. As requested, further information and the three references concerning the molecular profile of metastases in canine and feline medicine were added in lines 203-205.
Line 33: if it possible, the age of the sterilization should be reported, since it could add an interesting data, in particular concerning the reactivity to ER and PR IHC. Dear reviewer, unfortunately, there is no clinical data regarding the age of the sterilization.
To confirm what stated in lines 80-81, the authors should add a photo of the anti-CK IHC of vessels, to show that the cells are actually neoplastic epithelial cells. If not possible, the authors should provide a photo of HE at higher magnification. As requested, a new photo of HE at higher magnification (400x) was added to Figure 2 (Figure 2A).
It could be interesting to have also a photo of the HE of distant metastases. Dear reviewer, two new photos of distant metastases were added in Figure 2 (Figures 2E and F).
Line 103: please clarify which method the authors used to count the Ki67 index, since in literature there are several methods. As requested, the used methodology was added in lines 120 and 121, as well as one reference.
I suggest to delete the part concerning the description of all the antibodies used in the study, since the table is very clear and this part is thus redundant. As suggested, the description of all the antibodies was removed from the main text.
An important point, lines 188-190. The authors stated that the exclusion from the differential diagnosis of the inflammatory carcinoma was also based on the COX2 weak staining. Based on my experience in COX2 and mammary cancer studies, I think that there is a common misunderstanding about the inflammatory pathogenesis of the inflammatory carcinoma, also due to its name. actually, this name comes from the macroscopic aspect (caused by lymphatic vessels massive embolization), and not to any concurrent inflammation. For this reason, even if previous articles showed a strong positivity to anti-COX2 IHC, it does not mean that COX2 is a marker of inflammatory cancer. In my opinion, this finding is not needed to discriminate between and anaplastic from an inflammatory carcinoma, I think that what you have described about macro and microscopic findings is sufficient to support your diagnosis. Dear reviewer, thank you for raising this very interesting point. As suggested, we rephrased the statement in lines 212-215.
Conclusion: the authors built this section only on the utility of the histological and IHC analyses in diagnose this disease. I suggest to emphasize also the diagnosis in its own, since it concerns a very rare (for the moment) and aggressive cancer in the feline species. As requested, additional information about the diagnosis of AMC was added in the conclusion section (lines 220-228).
Reviewer 3 Report
Dear authors,
The present work is very interesting for the medical community, however I advise you to better clarify certain points in the description of the clinical component of this case report.
In the clinical history, it is necessary to know the history of vaccinations, deworming and serological tests for FeLV and FIV.
Is there any relationship between the gastroenterological clinical signs and the patient's oncological situation?
It is important to know if the animal was sterilized or not and in this case to know if he was taking contraceptives or not.
In the field of hematological analysis, knowledge of the coagulation tests is important because prolonged coagulation times may be associated to disseminated intravascular coagulation in animals with advanced and metastazide mammary carcinoma.
The tumor's staging description is important, because in order to plan the adequate therapeutic approach and prognosis, it is essential to determine the local extent and the degree of spread throughout the body.
For tumor staging it would have been necessary to perform fine needle aspiration cytology or excisional biopsy of regional lymph nodes (axillary and inguinal) to assess possible regional metastization.
The histological grading of the feline MT was also not described.
Introduction
The authors could take advantage of the introduction space to briefly describe the theme of feline mammary tumors, namely malignant ones.
This space should be used to refer to everything that is known about anaplastic mammary carcinomas of the cat, namely the aforementioned work by Della alda et al.
References
References are not numbered
The better separation of the various themes during the description of the case report would facilitate the reading and make this article more appealing.
Good work
Author Response
Reviewer #3 (Comments to the Author):
Dear authors,
The present work is very interesting for the medical community, however I advise you to better clarify certain points in the description of the clinical component of this case report. Dear reviewer, thank you so much for your positive opinion about this case report, we appreciate your very valuable review.
In the clinical history, it is necessary to know the history of vaccinations, deworming and serological tests for FeLV and FIV. Dear reviewer, in lines 45-46 was added that the animal had vaccination and deworming overdue. However, there is no information regarding the FeLV or FIV status.
Is there any relationship between the gastroenterological clinical signs and the patient's oncological situation? No, we don’t think so. The animal was diagnosed with feline panleukopenia, once it cohabited with another cat that had been diagnosticated with feline panleukopenia.
It is important to know if the animal was sterilized or not and in this case to know if he was taking contraceptives or not. Dear reviewer, in line 45, is stated that the cat was spayed, however, there is no information regarding the administration of contraceptives.
In the field of hematological analysis, knowledge of the coagulation tests is important because prolonged coagulation times may be associated to disseminated intravascular coagulation in animals with advanced and metastazide mammary carcinoma. Dear reviewer, although we agree with your comment, unfortunately, it was not possible to perform the coagulation tests due to financial limitations.
The tumor's staging description is important, because in order to plan the adequate therapeutic approach and prognosis, it is essential to determine the local extent and the degree of spread throughout the body. For tumor staging it would have been necessary to perform fine needle aspiration cytology or excisional biopsy of regional lymph nodes (axillary and inguinal) to assess possible regional metastization. The histological grading of the feline MT was also not described. Dear reviewer, unfortunately, it was not possible to determine the initial tumor staging, because the owner doesn’t authorize the PAAF in regional lymph nodes.
Introduction
The authors could take advantage of the introduction space to briefly describe the theme of feline mammary tumors, namely malignant ones.
This space should be used to refer to everything that is known about anaplastic mammary carcinomas of the cat, namely the aforementioned work by Della alda et al. Dear reviewer, further information regarding feline mammary tumors and AMC was added in lines 29-32 and 35-39, as well as three references.
References
References are not numbered. Corrected.
The better separation of the various themes during the description of the case report would facilitate the reading and make this article more appealing. As suggested, the case report was divided into several separated parts: Introduction, Case Presentation, Material and Methods, Results, Discussion and Conclusion.